# Urban Cat Management in Australia—Evidence-Based Strategies for Success

**DOI:** 10.3390/ani15081083

**Published:** 2025-04-09

**Authors:** Jennifer Cotterell, Jacquie Rand, Rebekah Scotney

**Affiliations:** 1Australian Pet Welfare Foundation, Kenmore, QLD 4064, Australia; 2School of Veterinary Science, The University of Queensland, Gatton, QLD 4343, Australia; j.rand@uq.edu.au (J.R.); rebekah.scotney@uq.edu.au (R.S.)

**Keywords:** urban cat management, free-roaming cats, enforcement, assistive approach, community engagement, cat sterilization, human well-being, environment, wildlife, one welfare

## Abstract

Free-roaming cats in urban areas are a persistent challenge. Traditional management strategies, such as containment laws, impounding, and fines, have proven ineffective, particularly in low-income areas, where most free-roaming cats are found. Some are unidentified owned cats, but many are stray cats being cared for by semi-owners—community members who care for them without formally adopting them. Financial barriers to sterilization and cat containment in these communities contribute to unplanned litters, the maintenance of free-roaming cat populations, and continuing complaints. This paper explores the limitations of enforcement-based cat management through the lens of the One Welfare framework, underscoring the holistic benefits of an assistive approach. Offering free cat sterilization, microchipping, and registration services to owners and semi-owners, especially in disadvantaged areas, promotes a more effective, humane solution that advances animal welfare while addressing social and community well-being and decreasing the risk to wildlife. Such programs have significantly reduced the numbers of cats impounded and euthanized, lowered cat-related complaints, enhanced cat welfare, and strengthened trust and cooperation between authorities and communities. Legislative changes are required to optimize the effectiveness of these programs.

## 1. Introduction

Urban free-roaming cats present persistent challenges for local governments, animal welfare organizations, and communities. Traditional enforcement-based management strategies, such as containment laws, cat impoundment, and fines, have proven largely ineffective, particularly in low-income areas, where financial barriers limit access to sterilization and other preventive measures, including containment. This manuscript critically examines the challenges of punitive approaches to urban cat management, highlighting the interconnected factors that contribute to free-roaming cat populations, including the legal classification of cats, financial constraints on sterilization, and the ethical and emotional toll of euthanasia on veterinarians and animal management officers. Using the One Welfare framework [1], which acknowledges the interdependence of animal welfare, human well-being, and environmental sustainability, this paper explores the success of assistive Community Cat Programs that provide free sterilization, microchipping, and registration services. The effectiveness of such programs is demonstrated through case studies from Victoria, Queensland, and New South Wales, where Community Cat Programs have led to significant reductions in cat impoundments, euthanasia, and nuisance complaints [2,3,4].

This manuscript further discusses the broader implications of cat management strategies on veterinary mental health and job satisfaction [5,6,7]. High euthanasia rates, exacerbated by legislative definitions that classify semi-owned and unowned cats living around humans as feral [8], contribute to moral distress and burnout among veterinarians [9,10,11,12], animal shelter staff, and council officers. Additionally, the legal and regulatory frameworks governing cat management are examined, highlighting the challenges faced in managing cats in urban areas and the impact of financial barriers to responsible pet ownership. By presenting evidence-based recommendations for sustainable, humane cat management policies, this paper advocates for legislative reforms and long-term investment in preventive programs. These strategies not only improve outcomes for cats, but also reduce the emotional burden on veterinary professionals, enhance public trust in animal welfare authorities, and align urban pet management practices with contemporary community expectations. By decreasing free-roaming cat numbers and increasing the proportion of older sterilized cats, it will be possible to decrease the risk of predation and protect the environment [13]. By integrating lived experience with empirical evidence, the authors present a comprehensive analysis that bridges research with real-world application, reinforcing the validity and relevance of their recommendations. This firsthand knowledge ensures that the proposed solutions are not only theoretically viable but also operationally feasible within the constraints of local government and animal welfare frameworks.

## 2. Challenges with Free-Roaming Cats in Urban Areas

Free-roaming cats in urban and peri-urban areas frequently generate complaints to authorities regarding soiling, property damage, noise disturbances, perceived risks to wildlife populations, and potential disease risks to humans and animals [14,15,16]. Unsterilized and free-roaming cats often cause nuisance complaints, particularly during breeding seasons, leading to trapping, impoundment, and possible euthanasia [4]. Most free-roaming cats in Australian urban areas are unidentifiable, lacking microchips or collars, and are classified as “stray” in shelter records [17]. Reclaim rates for these cats are low, averaging 5% [17]. Some of these “stray” cats are unidentified owned cats, but many are cats being cared for by compassionate community members, known as semi-owners, who provide food and basic care but do not perceive them as their property [18].

In Australia, 3–9% of adults feed one or more cats daily, with an average of 1.5 cats per caregiver [19,20,21]. Most of these cats are unsterilized, contributing to unwanted litters, and are often timid and fearful of unfamiliar people, making them more likely to be euthanized if impounded [22]. Free-roaming cats are especially prevalent in low socioeconomic areas, where financial pressures, exacerbated by rising living costs, limit access to veterinary services and compliance with regulations like containment, microchipping, and registration for owned cats [23,24,25,26,27].

Providing support for owners and semi-owners, such as subsidized sterilization, microchipping, and registration programs for their cats, and information indicating that cats can be pregnant as early as 4 months of age, can break the breeding cycle and improve welfare outcomes in disadvantaged communities [2,4,28]. However, many subsidized programs remain unaffordable for low-income communities, perpetuating free-roaming cat populations and associated issues. Redirecting resources toward accessible sterilization and microchipping programs is essential for effective population management and improved outcomes for both cats and communities [21,29].

This manuscript establishes a foundational framework for effective domestic cat management by adopting the classification system recommended by the Royal Society for the Prevention of Cruelty to Animals (RSPCA) Australia [8], clearly distinguishing between feral cats and domestic cats, which are further subdivided into owned, semi-owned (cats fed or provided with other care by people who do not consider that they own them), or unowned (receiving food unintentionally from humans). These distinctions are critical in shaping evidence-based management strategies, as they recognize the varying degrees of human involvement in cat populations, which has implications for effective approaches to intervention. By clearly defining these categories, this manuscript provides a basis for the implementation of targeted, humane, and sustainable cat management programs that address nuisance concerns, reduce euthanasia, improve welfare outcomes, and reduce the risks to wildlife [2,3,4]. This classification framework also highlights the limitations of defining free-roaming cats in urban and peri-urban areas as “feral”, which restricts the management options to ineffective lethal control measures [8]. Instead, the proposed terminology supports more nuanced and effective solutions, such as Community Cat Programs and high-volume sterilization, which have demonstrated success in stabilizing and reducing urban cat populations and the issues that they cause [2,4,30,31].

For the successful management of cats in the urban and peri-urban areas of Australia, as well as around farm buildings and in indigenous communities, it is critically important that semi-owned cats are not considered feral cats, as proposed in the federal government’s Threat Abatement Plan [32]. A key point of contention in urban cat management is the differing criteria for success between stakeholders, which are shaped by their respective underlying values, priorities, and beliefs. Our manuscript evaluates management strategies based on metrics such as reductions in cat impoundments, euthanasia, complaints, and costs and improvements in community trust and welfare. However, others advocate for the complete removal of free-roaming cats through stringent enforcement measures, including 24/7 containment and impoundment policies, with the humane killing of cats not adopted. Acknowledging this divergence is crucial, as it highlights why enforcement-driven approaches, while aligned with the goal of eliminating free-roaming cats, have consistently failed to achieve the long-term resolution of issues associated with free-roaming urban cats. For example, despite introducing 24/7 cat containment measures (curfews), the Yarra Ranges and Casey Councils in Victoria have not achieved a reduction in free-roaming cats or associated complaints. In Yarra Ranges, three years after the cat curfew was implemented, cat complaints had increased by 143%, impoundments by 68%, and euthanasia rates by 18%, despite minimal human population growth over the same period. Similarly, in Casey, even after 20 years of mandated cat containment, intake remained 296% higher than the baseline levels, and nuisance complaints showed no meaningful reduction. These outcomes illustrate that containment laws alone are ineffective and, in fact, can exacerbate cat-related issues by driving increases in complaints and impoundments and placing additional strain on council resources [33,34,35]. These experiences from councils reflect the findings in the RSPCA’s report Identifying Best Practice Domestic Cat Management in Australia (2018), which highlights that councils implementing mandatory cat containment have not demonstrated measurable reductions in cat complaints or roaming cats [8]. The RSPCA also acknowledges that containment laws fail to address the core issue of semi-owned cats—the primary source of free-roaming cats—and that such laws can increase impoundments, euthanasia, and community tension without delivering long-term solutions. Empirical evidence from Australia and internationally demonstrates that enforcement—without complementary, assistive-based strategies—results in persistent population turnover, community non-compliance, and high euthanasia rates without effectively reducing complaints associated with free-roaming cats.

## 3. Changes in Urban Cat Management

In many Australian states, the responsibility for urban cat management by local governments has only been legislated in the last 20 to 30 years. For example, in Victoria, prior to the enactment of the Victorian Domestic Animal Act 1994 [36], with only the Dog Act in place, there was no legal requirement for cat management in local government areas. The Domestic Animal Act 1994 set out legislative requirements for cats to be registered with the council that they resided in and wear a visible identification tag provided by the local government (council). From 1994, management efforts were primarily focused on promoting responsible cat ownership to residents, including providing literature on the keeping of cats, cat health and welfare, and the importance of sterilization, as well as legislative requirements relating to registration and containment. Simultaneously, trapping and impoundment programs for nuisance cats were implemented.

The animal management service provider for the city of Banyule, Victoria (human population 127,370 in 2021), which is part of Greater Melbourne (human population 4,917,750 in 2021) [37,38], is the Cat Protection Society (CPS). In 2011, the CPS had multiple council contracts, which resulted in their facility being at capacity for most of the year and beyond capacity through kitten season (September–March). Operating above capacity to provide adequate care for cats led to high euthanasia rates, with a reported 90% of the 12,000 cats admitted in 2011 being euthanized [39,40]. During peak kitten season, kittens less than 8 weeks of age were commonly euthanized on admission due to a lack of available foster homes to provide care until the kittens reached an adoptable age. After years of exposure to transporting numerous healthy cats and kittens to be euthanized at the CPS, the psychological harm experienced by two AMOs led to a call for substantial change in cat management practices in 2012 (personal experience of J.C.). Specifically, this was precipitated by the death of yet another healthy, socialized kitten, which was euthanized on admission due to a lack of capacity at the shelter.

Regulatory measures such as cat ownership limits, curfews, and the prohibition of trap–neuter–return (TNR) and return-to-field (RTF) have proven ineffective in addressing cat overpopulation in urban environments. Regulatory policies often fail to achieve sustainable reductions in free-roaming cat populations and the issues that they cause and may instead contribute to unintended consequences, including increased abandonment, diminished community participation in management efforts, a rise in nuisance complaints, higher euthanasia, and negative impacts on the job satisfaction and mental health of involved staff [4,33,34,35]. In contrast, evidence-based approaches that incorporate sterilization, community-supported management, and non-lethal population control strategies have been shown to be more effective in stabilizing and reducing free-roaming cat numbers while improving human and animal welfare outcomes [2,3,4,41,42].

In 2013, a two-pronged approach was implemented that shifted the cat management paradigm in the city of Banyule towards a more humane approach. Firstly, AMOs embraced an assistive style. This focused on helping cat owners and cat carers to better care for their cats, including, where possible, taking ownership of stray cats being cared for, as well as working to resolve complaint issues, rather than simply impounding nuisance cats. Secondly, a free cat sterilization program was implemented to stop unwanted kittens being born. The three suburbs with high cat-related calls and impoundments were targeted, with further microtargeting within these suburbs to locations of cat-related nuisance calls. Some microtargeting also occurred in problem locations across the whole city [4].

## 4. Summarizing a New Way of Urban Cat Management

Banyule’s AMOs developed a strategy centered around free sterilization, microchipping, and registration for owned and semi-owned cats, specifically targeting locations with high cat-related complaints, which were also identified as low socioeconomic areas. This contemporary approach focused on resolving complaints and reducing cat admissions to the contracted shelter through sterilization, microchipping, and registration, instead of impounding and applying infringements. Trap cages continued to be provided to residents to trap nuisance cats. However, where possible, AMOs engaged with residents to identify cat owners and carers whose cats were a source of nuisance complaints, to understand the situation with their cats and their management challenges and negotiate positive outcomes. This approach was not designed to replace enforcement, but was a first option to resolve animal welfare, nuisance, and cat overpopulation issues [43]. When owners were identified, free cat sterilization was offered and a seven-day window afforded them to address the reported problems before the complainant was contacted to check if the issue was resolved or persisted. Trapped identified cats were returned to their owners without impoundment where possible, and efforts were made to reunite impounded cats using internal databases. This comprehensive approach aimed to reduce the number of cats entering the pound system and being euthanized and mitigate the subsequent negative impact on staff by addressing the root causes of free-roaming cat populations. This program greatly reduced cat impoundments, euthanasia, and complaints [4].

Recently, two other studies from sites in major cities and rural locations in New South Wales (NSW) and Queensland (Qld) have reported comparable outcomes with similar Community Cat Programs based on assisting cat owners to care for their cats, encouraging semi-owners to take ownership, and providing free cat sterilization for owned and semi-owned cats, targeted and microtargeted to problem areas. Collectively, these reports from three states in Australia, with differing legislation relating to urban cats, reinforce that an assistive rather than an enforcement approach, coupled with targeted free cat sterilization programs, is a highly effective strategy for urban cat management [2,3,4,44,45]. This assistive approach is aligned with the One Welfare philosophy, based on the interconnectedness of human, animal, and environmental well-being, and research demonstrating that improving animal well-being will benefit humans and their physical and social environments [1].

### 4.1. City of Banyule Program

A Community Cat Program was first introduced in 2013 in the city of Banyule. This initiative integrated a targeted sterilization, microchipping, and registration (licensing) program with an assistive, rather than punitive, approach to managing cat-related issues. The program was solely operated by AMOs, who focused on working with the community to help cat owners and carers to sterilize their cats. It was funded by the council and assisted by supportive veterinarians, who provided cat sterilization surgeries at a reduced cost to the council. Semi-owned cats were required to be microchipped and registered to an owner, so there was a contactable person responsible for the cat. This model highlights the effectiveness of an AMO-driven, collaborative, and assistive management program in stabilizing and reducing free-roaming cat populations and associated issues while promoting improved welfare outcomes. This approach required exceptional community engagement skills, as the AMOs needed complete honesty to assess the extent of the issue—which varied in terms of the cat population size, differing in each property—while ensuring that community members did not fear penalties or legal repercussions for situations such as having excess, unconfined, or unregistered cats. The success of this approach relied on support from the council. This was garnered by addressing three aspects: providing councilors and council staff with evidence-based advice on the effective management of cats; demonstrating the efficacy and long-term benefits of the program to budget decision-makers in the council; and ensuring that customer service staff were equipped to engage with the community and help deliver the program. It also involved strengthening collaboration with other agencies and social services. The AMOs established processes that targeted the underlying causes of complaint calls. Rather than relying on reactive trapping programs as a first response, the strategy prioritized an assistive approach with targeted interventions that addressed the underlying factors contributing to cat-related complaints and overpopulation.

### 4.2. City of Ipswich, Queensland—Australian Pet Welfare Foundation

The second program, in Ipswich, Queensland (Qld), was initiated in 2020 by the Australian Pet Welfare Foundation [2], a research and advocacy organization, in collaboration with RSPCA Queensland, Australia and later with the Animal Welfare League (AWL) Qld. It is largely funded by donations, including from the Foundation Brigette Bardot (Paris, France), vaccines and parasite treatment from MSD Animal Health (Macquarie Park, NSW, Australia), and free microchip registration by Central Animal Records (Keysborough), with a small amount of state and local government funding. Support from the city of Ipswich was limited, and community engagement was provided by staff from the three welfare organizations involved. Trap cages continued to be provided to residents, and AMOs continued to trap cats at problematic sites. Over time, the support from the animal management team increased after the benefits of the program became apparent and complaints from residents did not ensue. Owned cats were sterilized and, where possible, semi-owned cats became owned by the carer. Where there were multiple semi-owned cats at a site, in most cases, after sterilization and microchipping, they were returned to their outdoor homes for continued care by the semi-owner, rather than being microchipped as owned cats. In the city of Ipswich, cats are not required to be registered, but owners with more than two cats are required to apply for an excess cat permit costing A$520 and cats must be confined to the owner’s property. A research permit provided by the Queensland Department of Fisheries and Agriculture (DAF) [46] allowed semi-owned and unowned cats to be sterilized, microchipped, and cared for without the requirement to become owned. This mostly involved situations where multiple cats were being cared for, as well as a small number of healthy stray cats that were impounded but found not to be adoptable because of fearful behavior towards humans. These cats were subsequently returned home following sterilization and microchipping (RTF). The majority of the sterilized cats were owned (74%) or semi-owned cats that became fully owned (10%), while 11% remained semi-owned at multi-cat sites cared for by semi-owners. A small percentage (0.6%) were returned to the field. Unowned domestic cats, defined as those relying solely on unintentional human food sources (e.g., garbage bins), were not observed, likely because compassionate community members often began providing food. Under the Queensland research permit, if a location with multiple unowned cats was identified, the protocol was to sterilize and microchip the cats and appoint a carer responsible for feeding and monitoring their health, thereby transforming them into semi-owned cats.

An assistive approach was also employed by the staff of the animal welfare organizations to resolve cat-related complaints and reduce the risk of offending cats being trapped and impounded. This included providing motion-detecting cameras to obtain images of trespassing cats, letterbox flyers of images of offending cats with offers of free cat sterilization, and the provision of deterrents such as motion-detecting water sprays or helping to construct containment fencing. Typically, a combination of one or more of these methods was effective in resolving nuisance issues.

### 4.3. RSPCA New South Wales—“Keeping Cats Safe at Home”

The third program was initiated in 2021 by RSPCA New South Wales (NSW) in partnership with eleven councils as part of the project “Keeping Cats Safe at Home”, which was funded by the NSW Government through its Environmental Trust [3]. In some partner councils, AMOs provided support with community engagement, as well as trapping and transporting cats to and from sterilization appointments. The program offered free sterilization and microchipping for cats and was targeted towards caregivers of stray cats (“semi-owners”) and cat owners overwhelmed with multiple cats. Each participating cat was required to have a person designated as their “owner” and thus responsible for their ongoing care. The requirement to pay a lifetime registration fee for all cats who were microchipped in NSW was identified as an important barrier to participation in the program because it was not funded by the program, in contrast to the city of Banyule program, where the cost was waived. Hence, while microchipping was encouraged and offered free of charge at the time of sterilization, it remained optional. There was concern that information in the state microchip database could potentially be used for enforcement actions against the owners of unregistered cats. Cat registration in NSW costs A$68 (A$34 for pensioners) for lifetime registration, and there is an additional cost of A$96 for an annual breeder permit if the cat was sterilized after 4 months old—even if it was adopted when it was older than 4 months [47,48]. A late fee of a further A$22 is also levied if the annual permit is not paid within 28 days. These expenses can be prohibitive, especially for those managing multiple cats, leading to a reluctance to microchip in some situations.

## 5. Beneficiaries of the New Assistive-Based Cat Management

### 5.1. Impacts on Cat Impoundments and Euthanasia

The impacts of the three Community Cat Programs have been previously described and reported in other studies and reports; however, this summary focuses on the key beneficiaries of these programs. This analysis synthesizes previously reported findings to highlight the outcomes for cats, caregivers, and the broader community, demonstrating the effectiveness of targeted, community-based management strategies.

We have previously reported that, over the 8 years of the program being operated in the City of Banyule (financial years 2013/14 to 2021/22), cat impoundments decreased city-wide by 66%, euthanasia decreased by 82%, the proportion of cats reclaimed by owners increased by 62%, and cat-related calls to the council decreased by 36%, and a total of 831 cats were sterilized, microchipped, and registered with the council [4]. The number of kittens under 12 weeks of age (legal age for registration) that were impounded decreased by 75%. Over the final four years of the program, when data were available (ending 2021), the number of cats and kittens surrendered by owners to the shelter decreased by 50%, and the number of stray cats found trespassing on private property and directly handed into the shelter by residents decreased by 28% [4].

In Queensland, after a pilot program in a rural town (population 3000) in the City of Ipswich, cat intake into the receiving shelters was reduced by 60% and euthanasia by 85% in the third year of the Community Cat Program [2]. Cat-related calls also decreased by 39%. In the NSW program, the City of Paramatta (population 256,000), which is part of Greater Sydney, reported a 46% reduction in the intake of cats and kittens to the RSPCA Sydney shelter, a 41% reduction in cats arriving at the council pound, and a 49% reduction in cat-related nuisance complaints to the council [3]. In two small country towns (Weddin: human population 3608 and Walgett: human population 5250), cat-related nuisance complaints to the council decreased by 66% and 91%, respectively [3], after one year, indicating that these programs had significant and rapid impacts in reducing issues associated with roaming cats and unwanted litters of kittens.

### 5.2. Benefits to Cats and the People Who Care for Them

The greatest benefit of this assistive approach incorporating a free cat sterilization program is cat welfare and the well-being of the people caring for cats. Humans benefit from an assistive rather than an enforcement approach because cat owners and semi-owners have an emotional attachment to the cats that they are caring for and can suffer psychological injury when the cats are killed [13,49,50,51]. The human–animal bond has many social and psychological benefits for both the people and the animal [51,52,53]; therefore, facilitating and supporting this bond is imperative to the health and well-being of both.

Animal welfare was also improved, because cats being fed and provided with some form of care in the community, once enrolled in the program, mostly became fully owned companion animals. People who had been providing care for stray cats were encouraged to take full responsibility and accept ownership. In the Queensland program, under the DAF research permit [46], carers of multiple cats continued to provide care for their cats after sterilization and microchipping, but the carers were not required to be listed in the microchip database as the owners; instead, the mobile phone numbers of the organization operating the program (Australian Pet Welfare Foundation) were listed as a secondary contact, including the personal mobile number of one of the authors, who is a veterinarian (J.R.). This facilitated rapid decision-making in the event of the injury or sickness of a cat, and an internal database accessible 24/7 provided the contact details of the carer and the home location of the cat. Sterilization programs benefit cats in other ways, including reduced fighting, reduced disease risks, and a lower likelihood of nuisance complaints [2,3,4], which ultimately decreases the risk of the cat being trapped and potentially euthanized.

Where the number of cats at a property raises welfare concerns for the cats or the caregiver, offering the option for a shelter or rescue group to assist with rehoming some of the cats can reduce the physical and financial burden on the owner. Following sterilization, cats often become more affectionate and calmer [13], which increases the potential for rehoming over time. However, without trust and honesty, an AMO will not be able to address the initial extent of an issue or assist in supportive approaches to management and rehoming.

Subjectively, AMOs observed that the sociability of impounded and surrendered cats and kittens improved over the eight years of the Banyule program. AMOs were very rarely impounding kittens being born under houses, around factories, and at overwhelmed semi-owners’ properties. The AMOs’ personal observation (author J.C.) was that they were more frequently picking up kittens from accidental litters that were reared in homes and were handled. The increased sociability of the kittens potentially also contributed to decreasing the length of stay and the associated shelter costs for rehoming, increased the rehoming possibility, and contributed to a decrease in the number of cats and kittens euthanized.

### 5.3. Benefits for AMOs and Animal Care Workers

The traditional enforcement of domestic cat legislation can have significant mental health impacts on AMOs, which can subsequently affect their physical health [4]. These impacts often stem from barriers to implementing effective preventive programs, such as insufficient resources, inadequate funding, and other constraints, alongside a predominant emphasis on enforcement [4,54]. This historical enforcement-based approach, and the lack of effective programs to prevent kitten births, results in high euthanasia rates and thus negatively impacts the job satisfaction and psychological well-being of AMOs. The euthanasia of animals is well recognized as causing negative emotional and mental health effects and negatively impacts job satisfaction [5,54,55,56], particularly when euthanasia is viewed as unnecessary [49,57,58,59,60,61]. With the introduction of an assistive and collaborative approach, combined with a free sterilization program for owned and semi-owned cats, there was a noticeable increase in job satisfaction and improved mental health in Banyule’s AMOs [4]. The author (J.C.) felt empowered, being able to provide solutions to the community for cat-related issues, rather than maintaining the status quo of disempowerment and an inability to effectively assist residents or their cats in living happy, healthy lives. Being able to provide non-lethal solutions for nuisance cats increased the rapport within the community, as well as leading to greater compliance, and provided mechanisms to increase cat welfare and human well-being. Organizational support, empowerment, job resources, and feeling able to control one’s work situation are significant contributors to job satisfaction and positive well-being [58,62].

### 5.4. Benefits for Council and Welfare Agencies

Benefits to city of Banyule included reduced cat-related calls and associated costs [4], as well as improved job satisfaction for front-line staff, i.e., the AMOs, which ultimately contributes to career longevity. In addition to financial savings, efficiencies were seen in staff time and task allocation, and there was greater community support for the council and its cat management program. For shelters, these programs also translate to reduced costs and enable staff to be redeployed to proactive programs to further provide outreach to the community, improve cat welfare, and improve job satisfaction.

Cat-related calls to city of Banyule decreased in the target area by 51% and city-wide by 36%. The savings associated with reduced cat-related calls to the council, and the reduction in time spent by AMOs addressing complaints, estimated at A$290 per call, amounted to approximately A$137,170 over the 8 years of the program [4]. Further savings to the council arose from reduced costs associated with charges from the contracted shelter (A$303,490). The total estimated savings over eight years were A$ 440,660 [4]. The cost to the council for sterilization and microchipping amounted to A$ 77,490 for the 8 years. Similar or greater decreases in cat-related calls to councils in NSW and Queensland over one to three years were reported following the implementation of similar programs [2].

The flow-on effects of these programs include significant benefits to the contracted shelter due to reduced cat intake. Given the estimated cost for each admitted cat of at least A$400, and, in some cases, costs of over A$1000 for housing, sterilization, microchipping, and miscellaneous veterinary care, it is estimated that the program in the City of Banyule saved the shelter contracted (CPS) approximately A$619,942 over the eight years based on a cost of A$400/cat, minus the reduced income associated with fewer impounded cats from Banyule (A$80/cat until 2017/18 and A$150/cat from 2018/2021) [4,63]. Over the eight years of the program, the total savings to the local government and the contracted shelter were estimated to be closer to A$1 million [63], which did not include the value of the social or environmental benefits.

The budget for the sterilization program was considerably less substantial when compared to the increasing costs of the traditional trap, reclaim, rehome, or euthanize methods currently utilized in Australia. Local councils often face limited resources and capacity when managing community cat populations, leading them to rely heavily on external service providers. It is important that the cost of the service be evaluated annually and areas for improvement be identified. This could include changes or updates to the best practice methods for the animals and the service needs of the community and/or identifying services relevant to the stakeholders involved—for example, partnering with rescue groups and/or other veterinary services.

### 5.5. Benefits to Cat Rescue Groups

When shelters and pounds reach full capacity, excess cats may be transferred to rescue groups, often with documented agreements that mandate sterilization prior to rehoming. Many rescue groups operate foster care networks composed of dedicated members of the public, who provide temporary housing and care for these cats [64,65,66,67]. The responsibility for covering the costs associated with housing, general care, and veterinary services, including vaccination, microchipping, and sterilization, lies with the rescue groups. This financial obligation can be substantial, with some groups spending up to A$200,000 or more annually on the cats under their care [68,69,70]. The Banyule program had a direct positive impact on local cat rescue groups and carers. For example, following sterilization, some cat owners decided that they had too many cats and kittens to care for and relinquished them to a rescue group, knowing that they would be rehomed and not euthanized. The financial burden and other resource costs for the sterilization and microchipping of the cats were removed because the council had already absorbed this cost. In recognition of the financial burden on rescue groups, the Victorian state government has established grant schemes that enable eligible rescue groups to apply for additional funding to assist with cat-related expenses [71].

### 5.6. Benefits for the Environment

Adjacent to the three targeted Banyule suburbs (Bellfield, Heidelberg Heights, Heidelberg West) that contributed to the council’s highest rates of cat intake and cat-related complaints was the Darebin Creek Parklands. This parkland encompasses a wetland ecosystem supporting a rich diversity of birds, mammals, reptiles, fish, frogs, and insects [72]. The implementation of a targeted free cat sterilization program in these three suburbs resulted in a 51% decrease in cat-related complaints and a 73% reduction in impoundments. These outcomes suggest a significant decline in the population of free-roaming cats within the parkland precinct, thereby fostering a safer environment for native wildlife. Across the city of Banyule, fewer free-roaming, unsterilized cats would be expected to benefit the environment by decreasing the likelihood of wildlife predation, particularly from younger cats, who are more prone to hunting [73].

Additionally, the social amenities improved in Banyule, with reduced cat-related complaints to the council concerning soiling and noise disturbances. The risk of disease transmission among pets, wildlife, and humans was also reduced. In pet cats, the prevalence of feline immunodeficiency virus (FIV), along with cat-fight abscesses and cellulitis, decreased with fewer free-roaming, unsterilized cats [74], while sexually intact adult cats with outdoor-access have a higher likelihood of FIV [75,76]. For wildlife, the risk of toxoplasmosis is lowered, as most environmental contamination with toxoplasma oocysts originates from cats under one year old; reducing kitten births thereby lowers the overall risk [77]. Although toxoplasmosis is mainly food-borne in humans, the presence of fewer young, free-roaming cats also contributes to decreased exposure, although reduced environmental contamination likely would have greater benefits for wildlife [78].

## 6. Critical Success Factors for Effective Urban Cat Management

Effective urban cat management requires a multi-faceted, evidence-based approach. Critical success factors include understanding the community and gaining their trust through recognizing and valuing the bonds between cat owners, semi-owners, and their animals. A fundamental shift from traditional enforcement-based strategies to a community-centered, assistive approach is essential. It is vital that council employees and AMOs engage with communities through targeted support and updated training in best practices. High-intensity, free sterilization programs must be strategically targeted and microtargeted to the areas of greatest need, with all known participation barriers removed to maximize the impact. State and local government laws urgently need amendment to maximize the effectiveness of urban cat management.

### 6.1. Critical Success Factors—The People

It is vital that the relationship that cat owners and carers have with their cats is recognized and valued. Utilizing this human–animal connectiveness facilitates effective long-term management and optimizes the well-being of animals, humans, and the environment. Disregarding humans and their attachment to the cats that they care for, overlooks the importance of the human–animal bond in effectively managing these cats. Because the bond between semi-owners and their cats is almost identical to the bond between owners and their cats [50], respecting the value of the bond that both owners and semi-owners have with their cats provides an opportunity for the effective management of cats in urban areas.

### 6.2. Understanding Community Needs

Before commencing a program, it is recommended that a thorough needs assessment of the community is conducted. This involves identifying the best locations for an initial program by understanding the community and its demographics and researching locations with the highest stray and owner-relinquished intake of cats and kittens or cat-related calls to the council. It is important to identify the level of disadvantage in the area and assess the availability of appropriate services, such as access to veterinary services, especially if any currently provide services for low-income cat owners. Additionally, potential partners should be identified, such as welfare agencies or rescue groups, who can help to address some of the challenges associated with owned and semi-owned cats in disadvantaged areas. This includes providing disadvantaged residents with access to free cat food, sterilization, and transport for surgery [79,80]. Investing in a well-researched and community-tailored program ensures a greater likelihood of success. When the program is targeted to locations with the greatest need and fosters trust, it is more likely to be successful. This approach improves animal welfare and creates long-term, sustainable solutions for both cats and their caregivers. Additionally, such programs deliver significant financial benefits by reducing cat intake and complaints, leading to lower sheltering and enforcement costs. By addressing the root causes and empowering the community, officers can allocate their time more proactively to an assistive approach based on community engagement, rather than reactive enforcement, maximizing the efficiency and outcomes for all stakeholders. Understanding community needs requires effective communication and trust.

The success of the programs reported from Victoria, NSW, and Queensland was reliant on the relationships, rapport, and empowerment of the community [2,3,4]. A study in Bristol, UK [81] concluded that it is paramount to have effective engagement strategies in place in low socioeconomic areas, as standard messaging and communications may prove to be ineffective in these communities due to barriers such as distrust of service providers, limited social support networks, and social isolation, exacerbated by limited or no access to the internet or smartphone technology. As a result, communities can become disengaged and categorized as ‘hard to reach’, leading to low participation rates in programs. The authors also suggested that community-based partnerships can lead to sustainable behavior change. In all three Australian programs, this behavior change manifested as the conversion of semi-owned stray cats into fully owned pets [2,3,4,81].

Successfully implementing domestic animal management and addressing animal welfare issues requires engagement with disadvantaged residents and effective communication to gain a deep understanding of the community and their needs [43]. In accordance with Arnstein’s ladder of citizen participation, participation depends on how the community views the interaction with the agency involved [82]. If the community feels manipulated, they will not participate, while, if AMOs inform or consult with the community, there will be a degree of participation. When AMOs develop partnerships with the community so that negotiations can take place to initiate problem-solving initiatives, and there is joint decision-making, this garners the greatest participation from the community [82]. Effective, relatable communication is crucial for the success of an urban cat management program. Without trust and the ability to relate to enforcement officers on a personal level, it is challenging to build a rapport and encourage honesty and openness about individual and community situations. Consequently, the issues persist. When there is complete honesty, without fear of retribution, problems can be effectively addressed with mediation [43].

### 6.3. Building Trust

An initial major barrier with the residents of the city of Banyule was a lack of trust, reflected in the hesitance of residents to enroll cats in the program because of previous experience with, or knowledge of, past enforcement-only approaches taken by AMOs. This approach was what the community came to expect. Some believed that if they enrolled multiple cats in this program, there would be infringements issued, or possibly even cats taken from them. It took persistent communication and time to build trust and for the program to gather momentum with cat owners and carers. Over time, it was confirmed to the community that the AMOs wished to work with them to resolve the cat-related issues in the community. Of substantial importance was the flow-on effect, where community members reached out to AMOs for assistance with all domestic animal-related inquiries and associated problems. AMOs became a go-to for advice and assistance. This ultimately resulted in residents contacting the AMOs to alert them to the number of cats that they were caring for, asking for help and receiving the help needed.

As an example, effective community engagement was demonstrated in response to a complaint about a property with seven cats. Upon visiting the property, the AMO engaged in a conversation with the owner, who initially claimed to have only two cats. By taking the time to communicate effectively, show genuine interest, and find common ground, the AMO was able to build a rapport. The AMO explained the available options upfront, stating, “I am not going to issue a fine, but will help you manage the situation. If you have more than two cats, we can get them all sterilized at no cost to you. This is how it works...” This approach led to the owner admitting that he had seven cats, allowing for a more effective resolution. Another example involved a resident who called to enroll two female cats, aged two years and 18 months, for sterilization. Given the age and sex of the cats, it was suspected that they might have already had litters. During the conversation with the owner, the AMO inquired about any kittens that might need sterilization. The owner then revealed that there were an additional eight kittens and two more adult cats on the property. As a result, the AMO arranged for the sterilization of all cats and provided cages for their transportation to the veterinary clinic. Where cat limits are enforced, this leads to a lack of honesty when more than the allowed number cats is present, due to the fear that they will be removed and an infringement served or that the owner will be required to apply for a costly excess cat permit, which may be denied, especially if their neighbors are not supportive.

### 6.4. Assistive Approach to Urban Cat Management

A critical factor in success is community engagement, ideally through council-employed AMOs but, alternatively, via engagement or outreach officers from one or more animal welfare organizations and/or rescue groups. AMOs, staff in shelters, and animal rescue groups all have extensive knowledge of locations associated with cat-related complaints or annual litters of relinquished kittens. Proactively engaging with cat owners and semi-owners who are struggling to provide the necessary care for their cats, including sterilizing, microchipping, and registering them, and helping them to achieve better welfare for their cats, will also improve human well-being. Equally essential is supporting residents in resolving cat-related nuisance issues and mediating until all parties reach a satisfactory resolution, before considering more formal actions, and this process is most effectively driven by AMOs [2,4].

An alternative model to an AMO-led program is the employment of community engagement or outreach officers by one or more welfare agencies or rescue groups, seeking to engage with the community to identify the locations most at risk of cats being impounded or surrendered and to work with owners and semi-owners to sterilize their cats. This model can also contribute to effectively reducing cat intake and euthanasia in the receiving shelters [2]. Furthermore, assisting residents to resolve cat-related complaints is critically important in reducing nuisance-related calls to the council and hence reducing the risk of the cat being trapped and euthanized. A welfare agency-driven model was initially implemented in a rural town in the city of Ipswich, Queensland, and, in the third year, it successfully reduced cat intake into the receiving shelters by 60%, euthanasia by 85%, and cat-related calls to the council by 39% [2]. A further advantage of this approach was that minimal initial distrust existed between the community and outreach officers that may have been present with AMOs. However, the authority to reassure the residents that AMOs would not necessarily impound their cats was lacking. This approach is also less effective if AMOs continue to work in silos providing trap cages to complainants or trapping offending nuisance cats. This highlights the importance of the local government working in alignment with welfare agencies towards a common goal to achieve the best outcome for residents, the cats that they are caring for, and the environment.

In the City of Banyule, a free cat relinquishment service was provided for any community member who wished to rehome cats. This involved calling the council, and an AMO would attend the property and pick up the cat/s, taking them to the pound. In 2013, this process was modified to a more assistive approach so that, when a resident contacted the council to relinquish a cat or cats, the AMO would follow up with the community member initially by phone to make an appointment in person. The AMO would attend the property and discuss potential workable solutions to avoid sending these pets to the pound. This approach demonstrated that, by providing people with options and support with information and resources, they were able to make more informed decisions. As a result of this engagement, many individuals decided to keep their animals rather than relinquish them.

### 6.5. Cultural and Behavioral Change in Animal Management Officers

The engagement of AMOs in an assistance-based approach is crucial to a successful urban cat management program that is resource-efficient. In a study from a municipal shelter in Florida, USA [83], cat management transitioned from an enforcement-focused approach to a community outreach and assistance model. During this transition, it was noted that AMOs were the most resistant to this shift in mindset. Those AMOs who were unable to adapt were reassigned to other roles. This study highlighted the necessity of changing animal control systems, processes, attitudes, and behaviors at all levels to successfully implement a community outreach approach [83]. This is less resource-efficient and potentially less effective in situations where AMOs are not supported by council management to engage in an assistive approach and instead continue as a first line of action to provide trap cages or trap cats in response to calls relating to wandering and nuisance cats. AMOs work within communities day to day, and they already have knowledge of the areas where cat-related issues occur; thus, they are at an advantage in being able to address these problem areas promptly.

It was evident from the AMOs’ perspective (comm., author J.C.) that assisting community members and their cats helped to disolve barriers, enhanced relationships, and built trust. Additionally, it assisted in changing the public perception of AMOs from that of enforcement officers to that of an assistive community resource that residents could rely on to help them. This assistive-centered approach to cat management is aligned with the increasing societal expectations of animal care and welfare and the importance and benefits of human–animal connectedness. This garners approval and social license from stakeholders because it demonstrates legitimacy and builds trust within the community, while increasing job satisfaction among AMOs [84].

### 6.6. Training of Animal Management Officers

The primary goal of domestic animal legislation and the role of AMOs should be to assist the community in voluntary compliance by providing the necessary tools, information, advice, and resources, resorting to enforcement only when necessary. AMOs have discretionary powers under the Local Government Act [85], and the traditional methods of trapping, impounding, reclaiming, rehoming, or euthanizing are increasingly unacceptable to the community and to the shelter and council staff involved. Many residents fear handing over cats to AMOs because of the risk of euthanasia, reflecting a growing concern for animal welfare, which is now critical to the social license to operate across many animal industries in Australia [86]. The goal of the city of Banyule program was to improve cat welfare and human well-being in the three most disadvantaged suburbs by ensuring that owned and semi-owned cats were sterilized, registered, and microchipped, with semi-owned cats becoming fully owned by the residents caring for them.

Despite the complexity of urban cat management, there is no formal qualification specific to cat management for AMOs, resulting in inconsistent knowledge and approaches to addressing free-roaming cat populations. The training for AMOs in Victoria and other states of Australia does not include information on contemporary cat management or science-based cat behavior in the curriculum, and it largely involves on-the-job training. Therefore, the same cat management training process is applied to nearly every animal management officer commencing in a domestic animal management role. It is recommended that all new AMOs undergo compulsory training in animal handling, evidence-based best practices in animal management, community engagement, and effective strategies for engagement with the public. Certification courses should also be developed by the state government, and this training should be provided for all existing and prospective AMOs. A notable example of training for AMOs is offered by the National Animal Care and Control Association (NACA) in Florida, USA. Their programs provide animal control-specific qualifications, including training in animal behavior, mental health first aid, and community engagement through initiatives such as community-centered programming [87]. These training programs not only better prepare AMOs for the challenges of fieldwork but also ensure their success by equipping them with practical strategies for effective engagement with community members. This approach paves the way for a more equitable and successful model of field services, fostering collaboration and mutual understanding between AMOs and the communities that they serve [88].

### 6.7. Engagement of Stakeholders Who Can Assist with Community Engagement

The engagement of relevant stakeholders increases the effectiveness of cat management. Mental health and domestic violence support groups have a profound impact on animal welfare. However, key agencies such as social services, the police, and RSPCA inspectors often operate in silos, despite frequently attending the same properties that require assistance. To foster partnerships in the Banyule program, the AMOs engaged with social workers at community centers within disadvantaged areas. It was soon realized that different issues were being addressed with the same residents by different community service groups. Counseling sessions for residents with social workers at the community center provided an opportunity for engagment with other stakeholders. If direct links to animals were identified, the social worker would, with the client’s consent, involve an AMO to discuss companion animal-related issues. With the client’s permission, the AMO would contact the client via phone to address the specific needs of the animals and, if unsterilized cats were present, enroll them in the sterilization program. This collective approach benefited not only the people but also the animals in their care.

Relationships were extended to include social housing officers across the council area, who collaborated with the AMOs when animal welfare issues were apparent at social housing properties during property visits, evictions, and deaths. This collaboration often involved the presence of police officers in cases where known offenders resided. The council implemented an eight-day free welfare hold and funded the housing and care of any dog or cat involved in a welfare call to AMOs from a case worker. If long-term care was required, the council coordinated with the case worker/s and other organizations to ensure the animals’ well-being. These partnerships facilitated the proactive involvement of AMOs with key social services and other organizations during community safety meetings. These meetings were convened regularly to address community needs and focus on specific welfare concerns for individuals and their animals. Furthermore, integration into emergency management planning and the initiation of discussions with the dedicated council committee were implemented to address the inclusion of pets in emergency responses. Consequently, the city of Banyule updated its emergency response plan to permit the inclusion of animals at evacuation relief centers, which had previously been prohibited. The aim of this emergency response plan for situtions such as flooding, fire, or severe storm damage was to prevent the relinquishment of pets or their abandoment and subsequent impoundment and possible euthanasia.

Through various discussions across the council, it became evident that internal council departments were also operating in isolation, with many departments addressing varied issues at the same properties, often at the same time. These issues pertained to the health and well-being of residents (including children), their animals, and associated public amenity concerns. The author (J.C.), an AMO, identified a more effective approach to building partnerships within council departments. She developed a hoarding and squalor project plan, sanctioned by the council’s Director of City Development. This project established a dedicated Hoarding and Squalor Committee with a central question: what is our duty of care and legal liability to assist these residents? This initiative enabled the council to break down internal barriers, facilitate a more cohesive approach to individual residents (or properties), maintain a register of affected properties, and document outcomes aimed at reducing risks and improving the health and well-being of both people and their pets.

By engaging all stakeholders comprehensively and ensuring awareness of the program, stakeholders were prompted to inquire about the presence of cats at properties and their sterilization status. This led to the establishment of a referral program, whereby other agencies and organizations would contact AMOs with details of residents whose cats required enrollment in the sterilization program. The AMOs would then directly follow up with these residents and arrange transportation for the cats if necessary. This was considered by the AMOs to be the best option for community welfare, cat health, and environmental biodiversity.

### 6.8. Critical Organizational Change—Support of Elected Officials and Staff

Support for the program at all levels of the organization—in this case, the council—is important in maximizing the outcomes and for the sustainability of the program. This often requires cultural change, leading to behavioral change. Knowledge of the limitations of the current programs for cat management and the benefits of the proposed programs is critical to successful cultural and behavioral change.

Commencing the program in 2013, there was an unwillingness within the ranks of Banyule Council to promote the program to only those residents in the low socioeconomic target area, because it was perceived by some to be discriminatory towards residents city-wide. This was an initial barrier to the program, but effective communication and explanation resulted in agreement by the senior communications officer that initiation within suburbs with the highest cat-related calls and cat impoundments was necessary. This endorsement and support of the aims of the program, and the subsequent assistance with promotion (including promotional coverage by a national TV station), enhanced the program’s success [89]. Raising organizational awareness, providing information, and promoting behavior change were essential components of the program. The purpose and goals of the initiative were communicated by AMOs to councilors (elected officials), all levels of management, and council staff. The program was presented at council meetings, individual team meetings (e.g., customer service), and committees with external representation, such as the Environment and Climate Action Advisory Committee. Once the organization understood the rationale behind targeting disadvantaged areas and the goals of the program, they became active supporters and advocates for the initiative.

The organizational structure and management changes within Banyule’s animal management services were barriers on occasion. The support for AMOs continuing the program varied; this support was dependent on knowledge and the importance of the program to management, along with understanding the program, its aims, and it benefits to the council. Many local governments have centralized decision-making, mainly involving higher levels of management within the organization. Although this creates greater demands for efficient problem solving in these positions, because higher-level positions are not at the forefront, they only have partial knowledge of the problem, yet they still make decisions on behalf of the organization. Often, staff in these positions have no background or training in animal management. If senior management does not recognize the value of a program or the reasons that enforcement-based cat management fails to achieve its goals, they are unlikely to allocate the necessary resources or financial support for its implementation. It is the lower-level staff—in this case, the AMOs—who work within the community day-to-day, who have a greater understanding of the problems faced by the community, potential solutions, and the best options for resolution. In comparison, councils operating by a decentralized system, empowering lower-level staff by giving them more authority and enabling issues to be resolved more quickly, result in improved job satisfaction and happier employees [90]. In this case, the free cat sterilization program was designed and implemented by the AMOs, with minimal intervention by higher-level management, except for budget allocation for the program to continue [4].

## 7. Factor for Success—Sterilization Program for Owned and Semi-Owned Cats

Most cats entering shelters and pounds are born in the preceding 12 months [22,91], highlighting the need to stop unwanted litters of kittens adding to the free-roaming cat population. In addition, many of the nuisance complaints are noise issues associated with unsterilized cats and spraying by tom cats. Therefore, the second most critical component of a successful cat management program is a targeted sterilization program that minimizes the barriers to participation.

### 7.1. Intensity of Microtargeting to Locations of Cat-Related Calls or Impounded Cats

For the program to be most effective in achieving measurable impacts, it is important to identify areas that have the highest cat-related calls to the council or cat impoundments per 1000 residents. The number of cats that can be sterilized each year will be based on the budget available and the costs that can be negotiated for sterilization surgeries and microchipping. Provided that the program is microtargeted to locations within targeted suburbs where impoundments or calls are most likely to arise, the size of the area targeted will be determined by the funds available to sterilize 5 to 10 cats/1000 residents per year. It will also be determined by the rapidity with which measurable improvements in complaints and impoundments are required, with higher-intensity programs achieving more rapid results [2,41,92]. In the City of Banyule, an average of 4.1 cats/1000 residents per year were sterilized over 8 years in the three target suburbs, totaling a human population of approximately 13,445 [4]. If microtargeting is not possible for specific locations of cat-related calls or impoundments, higher numbers of cats (approximately 30–60 cats/1000 residents per year) will need to be sterilized to achieve a measurable decrease in intake over 3 to 4 years [41]. This model was used successfully in the pilot Community Cat Program in a small rural town in Qld, where 28 cats/1000 residents were sterilized per year over 3.4 years [2]. For the program to be the most resource-efficient, it is strongly recommended that programs be targeted to the most problematic suburbs and microtargeted within these suburbs to the locations of cat-related calls, and/or alternatively to locations from which impounded and relinquished cats and kittens emerge.

To illustrate the impact on the numbers of cats that need to be sterilized, and therefore budgeted for, in a city with the size of Banyule (127,370 in 2021 census) [37], if not targeted or microtargeted, across the city, between 3787 and 7574 cats would need to be sterilized each year for 2–4 years [2,41]. In contrast, if only targeted to the three most problematic suburbs (population 13,445), only 403 to 807 cats per year would need sterilization. However, if microtargeting is employed for the three suburbs, only 67 to 134 cats per year would need to be sterilized per year for 2 to 4 years, with smaller numbers thereafter. If microtargeting were employed across the city, the number of cats to be sterilized per year would equate to 631 to 1262 cats [4,92]. Community Cat Programs must continue over the long term, but the numbers to be sterilized after 2 to 4 years will substantially decrease. Long-term management is required because of the mobility of residents, especially those renting, leading to new residents arriving each year, some with unsterilized cats or kittens, and because of the impossibility of sterilizing every female cat in the area.

### 7.2. Supportive Veterinary Clinics for Sterilization Surgeries and Postoperative Care

The availability of affordable and accessible veterinary services for sterilization is a fundamental necessity for assistive-based cat management. The availability of veterinarians who will perform sterilization surgeries in healthy kittens weighing as little as 0.8–1 kg and lactating and pregnant females is an advantage given the frequency of these cats in the summer. A high capacity reduces the unit price and increases the capacity, but, currently, this is usually only available at animal welfare shelters in Australia, and veterinary schools do not yet provide training in this technique for graduating students.

Initially, a mobile veterinary surgery staffed by RSPCA Victoria was utilized in the Banyule program. However, issues with local veterinarians being required to deal with postoperative concerns, particularly after hours and mostly related to sutures, caused challenges and friction with local veterinarians. Two veterinary practices that had relationships with the council for the provision of veterinary services then offered to be involved and provide the surgical capacity needed. However, after participating for two years, one of the private veterinary clinics withdrew from the program, citing difficulties with the clientele from the targeted low socioeconomic suburbs. They found these clients disparate from their regular clientele and faced challenges in dealing with the owners and semi-owners of the enrolled cats. In contrast, the other participating private veterinary clinic, which had a well-established clientele of rescue groups and regularly dealt with semi-owners and cat caregivers, continued with the program. Although the loss of the first clinic initially created resourcing issues, the Cat Protection Society was recruited to provide their clinic services. This proved more efficient because of its strategic location, ensuring that residents had access to veterinary services in both the north and south of the city.

In contrast, the Qld program initially utlized a private veterinary clinic in the small rural town for the pilot program, but, as the program expanded to larger areas, a high-volume sterilizing capacity was provided initially by RSPCA Queensland, with additional capacity later provided by AWL Qld, who both generously met the funding provided by Foundation Brigette Bardot of EUR 30 per cat, i.e., approximately A$46.89/cat. In the NSW program, private veterinary clinics and the RSPCA Sydney Veterinary Hospital were utilized.

The availability of veterinary service providers for sterilization capacity is an essential component for program success. This mixed model of service providers, involving private veterinary clinics and a welfare organization, reflected their proximity and support, and this will vary from geographic location to location. In more remote areas, veterinary clinics may be the only option, but residents may require the transportation of cats to a town with a veterinary clinic. An alternative solution is a mobile surgery operated by an animal welfare organization, as was utilized initially. A lack of postoperative support if any problems arise is a limitation of this model, if there are no supportive veterinary clinics in the vicinity.

In many situations, services provided to local governments by private veterinary clinics or shelter organizations and the costs of sterilization and rehoming outweigh the cost of euthanasia. Therefore, private veterinary clinics that offer lower-cost euthanasia services for healthy and treatable cats and kittens under contracts with local councils should avoid subsidizing these procedures. Doing so removes the incentive for councils to invest in sterilization programs, which are essential for sustainable population management. It is recommended that the full client price for euthanasia be charged for healthy cats and kittens, as well as the consideration of an additional staff mental health levy. This is because of the negative impact on the mental health and job satisfaction of staff. Instead, the sterilization costs should be subsidized to provide greater incentive for proactive sterilization programs that benefit the job satisfaction and mental health of veterinarians and their staff.

### 7.3. Minimizing Barriers to Cat Owners and Semi-Owners: Cost

Subsidized cat sterilization programs have been described as a “substitution” for people who were already planning to get their pet sterilized, suggesting that this simply provides a cheaper option [93]. However, having barriers or criteria, such as requiring that participants have a government-issued benefit card, does not assist all residents in low socioeconomic and disadvantaged areas, where the sterilization costs are often unaffordable, even when at a reduced price, and where both owned and semi-owned cats need to be sterilized [2].

A reduced-cost spay–neuter program in Massachusetts, USA [94] was implemented by a not-for-profit feline rescue over a two-year period. The study surveyed 1287 participating owners and reported that owner income and procedure costs were barriers to accessing the program. Specifically, the decision to participate in the reduced-cost program was directly related to the owner’s income and the cost of the sterilization procedure. In conclusion, full-cost sterilization services often present major financial barriers for low-income residents, and low-cost alternatives provide some relief but are still out of reach for those cat owners and semi-owners at the greatest risk of their cats being impounded or relinquished, often because they are caring for multiple cats. However, implementing targeted, free cat sterilization programs funded by state and local governments, with assistance from welfare agencies, can substantially increase the early sterilization rates, minimize delays in these procedures, and eliminate financial obstacles for individuals [95]. This underscores the essential need to eliminate any potential barriers in the design and implementation of sterilization programs to ensure accessibility and thereby improve the overall program effectiveness.

### 7.4. When Free Is Not Free: The Financial Barriers to Cat Sterilization

In January 2022, the AMO driving the city of Banyule program retired after 22 years of service, and other staff also resigned, resulting in the departure of key staff with extensive knowledge and background in animal welfare and domestic animal management. Although a free sterilization program was retained, the lack of understanding of the underlying cause of cat overpopulation problems resulted in cost barriers being re-introduced. Specifically, for carers and owners to access the free cat sterilizaton program, costs of over A$200 had to be first paid to microchip and register their cats to meet the eligibility criteria to participate. Specifically, the new program stipulated that, before receiving free cat sterilization services, cat owners and carers were required to register their cats with the council as unsterilized animals, incurring a fee of A$112.50 (compared to A$40.50 for a sterilized cat) [96]. Additionally, a legal requirement required obtaining and providing a source number at an additional cost of A$24.50. The source number was obtained by online application to Animal Welfare Victoria and was necessary for the microchipping of a dog or cat. In addition, residents needed resources to access the internet and a smartphone or computer to obtain a source number. In Victoria, in accordance with the current regulations, it is a legal requirement that any transfer of ownership of a dog or cat, or advertisement for sale, must include both the microchip number for each animal and a source number generated by the Pet Exchange Register [97]. Failure to comply constitutes an offense. Therefore, before being able to enroll a cat for free sterilization, an initial cost of between A$230 and A$330 was required, which included registration as an entire animal, the microchip implantation costs, and the provision of a source number, which was prohibitive for many residents. These expenses were substantially greater than the average cost for the microchipping and then registration of a sterilized cat.

Consequently, community participation in the program was minimal, and the effects became evident in a discussion paper released in May 2024 as part of a community consultation process [98]. The paper reported that only 21 free cat sterilization vouchers were issued in 2021/22, and just nine vouchers were issued in 2022/23. The data did not clarify whether the vouchers were actually utilized for sterilization. In comparison, in the prior free program without these financial barriers, in 2019/20, 90 cats were sterilized, and, in 2021/22, a further 73 cats were sterilized. Similarly, when cost barriers were implemented in the Queensland free Community Cat Program (namely, a fee of A$50 was implemented in an attempt to make funds available for longer), there was an immediate marked decrease in the numbers of cats enrolled in the sterilization program, presumably excluding those cats that were most at risk of having unwanted kittens or being impounded [2]. The numbers increased again when the cost barrier was removed. In one of these suburbs, the median individual weekly income was A$636 per week, whereas the cost of sterilizing and microchipping a female cat at a private veterinary practice is in the range of A$300–500. This emphasizes the importance of low-income residents, particularly those with more than one cat, having access to free sterilization and microchipping for cats if urban cat populations are to be controlled.

## 8. Sustainability

The importance of understanding the key components that made all three programs successful, i.e., minimizing accessiblity barriers and taking an assistive rather than an enforcement approach, is well illustrated by data compiled across the 8 years of the Banyule program up to 2021. The AMOs originally appointed had backgrounds in animal welfare and shelter work, with strong knowledge of animal behavior and handling. However, following their departure, these skills were not a requirement for subsequent appointments, resulting in a limited understanding of the community’s needs and the rationale behind the program’s approach. Subsequent staff turnover led to the loss of program-specific knowledge, and, although the council continued to promote a free cat sterilization program, new AMOs were appointed under a model that is increasingly adopted by many councils, where officers are required to manage all bylaws rather than specialize in animal management. As a result, traditionally trained, multi-skilled AMOs without specialist expertise in animal handling or community engagement reintroduced a enforcement-focused approach to cat management. The effect of this loss of key assistive-focused staff was reflected in the updated cat management statistics presented in a City of Banyule discussion paper, which reported a 140% increase in cat intake within two years of the end of the assistive-based approach, combined with the end of the no-financial-barriers sterilization program in December 2021. Specifically, the number of impounded cats rose from 124 in 2019/20 and 125 in 2020/21 to 324 in 2021/22 and 299 in 2022/23. This demonstrates that there was no clear understanding at all levels of local government of the underlying causes of free-roaming cats or an awareness of the critical success factors of the program. These became clearer with reflection and many discussions between the authors (J.C. and J.R.) while analysing data and documenting the results for publications from the city of Banyule and city of Ipswich programs in 2024 [2,4], as well as the consideration of data from the NSW sites [3]. This subsequent reflection and learning highlights the importance of evaluating a program’s effectiveness based on data analysis and making modifications as indicated. In conclusion, long-term cultural change is necessary throughout councils and associated organizations to ensure the longevity of an assistive approach for effective urban cat management.

## 9. Minimizing Barriers to Residents’ Access to the Program—The Need for Legislative Change

One of the few barriers to the assistive program in the City of Banyule was that all semi-owned cats that were sterilized had to become owned by the carer or relinquished. This is because TNR is illegal in all areas of Australia under legislation related to containment, abandonment, and biosecurity [31,99,100]. In addition, all sterilized cats were to remain on private property and could not be present on public property, and all residents caring for semi-owned cats were obliged to assume full responsibility for the cat’s care and well-being, including registering their ownership details on the cat’s microchip. When sterilization, microchipping, and registration were provided at no cost, semi-owners and community cat feeders were willing to assume ownership of the cats, even when multiple cats were involved, because they trusted the AMOs not to penalize them for breaches of the relevant bylaws—for example, for having excess cats. In some cases, these cat caregivers had been feeding the cats for years and regularly relinquishing unplanned litters of kittens. This arrangement was advantageous as it ensured that these cats were no longer breeding within the community and were under the care of a responsible individual.

Residents in the program were also not followed up to pay additional annual registration fees after the first year was waived. When Victorian councils waive registration fees, they still must pay the State Treasury A$4.16, whereas, in NSW, the full registration fee must be paid to the state government. In NSW, the initial registration fee for a cat is A$68, with an additional annual breeder permit fee of A$96 if the cat is not sterilized by four months of age [47,48]. This is a substantial additional cost to the providers of free sterilization programs if the registration fees for disadvantaged cat owners and semi-owners are also included in the program. In NSW, most participating cats were microchipped with their caregiver’s consent but not registered due to the cost. Where participants declined microchipping, this was generally due to concerns about future enforcement actions by the council. In situations where semi-owners are unable or unwilling to take ownership, it is typically where there are multiple cats and the semi-owner fears potential compliance actions against them if they are the owner. To protect the welfare of cats and prevent breeding, it is strongly recommended in these situations that the cats are also sterilized and microchipped and secondary contact details are recorded—for example, for the welfare agency or rescue group operating the program. This facilitates a responsible person being available to be contacted by a veterinarian, shelter, or rescue group to make decisions regarding the best welfare outcome for the cat in a timely manner, particularly if the cat is injured or sick. This was the model that was successfully used in Queensland, where the secondary contact was the Australian Pet Welfare Foundation and two after-hours mobile phone numbers were provided (one was for author J.R.). The semi-owner’s details were recorded in an internal database for the foundation, but not in the microchip database.

### 9.1. Cat Limits and Mandated Containment

Within the city of Banyule, council bylaws limited the number of cats on a property to two, and keeping more than two cats required a permit from the council. Acquiring this permit included the need to obtain neighbor consent and incurred an additional fee of A$50 annually [101], potentially creating an additional barrier for community members willing to assume full responsibility for these cats, especially because many also own cats [102]. Permits for owning more than two to four cats are mandated in most Australian states, with the fees ranging from A$50 to over A$500. In certain local government areas, the annual renewal fees exceed A$300, particularly for households with more than five cats [101,103]. Additionally, some councils impose supplementary charges; for example, in the city of Ipswich, residents must pay A$72 to replace a registered cat and an extra A$113 to add additional cats to a permit [103].

Bylaws relating to cat limits and containment were not enforced for participants in any of the programs, with the Qld program covered by a government research permit allowing multiple cats to remain semi-owned with a caregiver after sterilization and microchipping; in NSW, cats are not required to be contained at the owner’s property. The AMOs in the city of Banyule had discretion to obtain the best outcomes in cat management for residents, their cats, and the council. To facilitate the enrollment of some of the most at-risk cats in the program and assist owners in keeping them over the long term, excess cat permits were waived [4]. In Qld, excess cat limits continued to be enforced, so, where two cats were already present, additional semi-owned cats were microchipped under the Qld DAF permit as semi-owned cats.

It is strongly recommended that state and local bylaws are amended to facilitate effective urban cat management. Anti-nuisance and animal welfare laws should be utilized and strengthened if necessary, and mandated containment, registration, cat limits, source numbers, and other barriers for low-income families should be removed, recognizing that free-roaming cats in urban areas are largely a reflection of the socioeconomic challenges associated with containing and sterilizing cats. Focusing on an assistive approach to resolving complaint issues is very important and, in most cases, is successful, without having to resort to enforcement.

### 9.2. TNR and Return to Field

The limitations in the city of Banyule program were that cats could not remain semi-owned, and the program could not involve cats that resided in public areas unless they became owned and relocated to the owner’s property. In the USA, the sterilization of semi-owned cats, or TNR, is frequently used to manage free-roaming cats, and some of the cats may reside in public spaces or around businesses [41,92,104,105,106]. Increasingly, in the USA, stray cats that are brought to an animal shelter by the public or impounded by AMOs are sterilized and returned to where they were found if they are deemed healthy but not readily adoptable because of their behavior. Returning cats that are unlikely to be adopted to their home location (called return to field) is based on the premise that, if the cat is healthy, it will have a carer in the vicinity of where it was found [107]. It is more likely to be reunited with its owner or carer by returning it to where it was found, rather than by holding it in the shelter, where the return-to-owner rates are low [107]. Cats are 13 times more likely to be reunited with their owners by means other than a visit to the shelter by the owner [108]. Return to field (RTF) is very important in minimizing the exposure of staff to the negative mental health effects of caring for a cat to socialize it and then, sometime later, a decision being made that it is not adoptable within an agreed timeframe and must be euthanized. Anecdotally, return to field is being embraced by shelter staff because it avoids the mental trauma of having to euthanize healthy, less socialized cats and improves job satisfaction [2,109]. This is being evaluated as part of an investigation of the impact of a Community Cat Program based on the sterilization of owned, semi-owned, and unowned cats under the research permit issued by the Queensland Department of Agriculture and Fisheries (DAF) [2,46]. However, return to field is also illegal in Australia. It is recommended that legislative amendments be made to allow both TNR and RTF in light of the adverse mental health impacts of the euthanasia of healthy cats and kittens on shelter and pound staff and AMOs, as well as members of the community caring for cats [50,57,58,59,110]. When these programs are applied with sufficient intensity and duration, they decrease free-roaming cats, the numbers of cats euthanized, and cat-related complaints [41,92,106]. This is especially important if there are no threatened and endangered native species at risk of cat predation in the area.

## 10. Conclusions

Legislative and enforcement appoaches for cat management over the decades have not been successful in reducing the numbers of free-roaming cats and associated issues [111]. This is because they fail to address the underlying causes of free-ranging cats in urban areas and are based on the false notion that it is irresponsible cat owners who are the cause of the problem. This failure of urban cat management is, in part, because the issue of free-roaming cats is a socioeconomic one and low-income households are often unable to comply, and in part, because many of the free-roaming cats are stray cats fed by compassionate members of the community. Understanding this, the considerable resources expended annually on enforcement [4,112] could be better allocated toward a more assistive and cost-effective approach, which would yield better outcomes for both animal welfare and human well-being, as well as for wildlife. Consequently, alternative strategies for urban cat management are urgently needed.

Based on the City of Banyule’s results and two other recent Australian examples from NSW and Queensland, and the many international examples, local governments should adopt strategies that emphasize proactive engagement with the community and stakeholders [2,3,4,43,113]. This has wide-ranging benefits because of the interconnectedness between animal welfare, human well-being, and their physical and social environments.

Community Cat Programs that are aligned with a One Welfare philosophy significantly reduce the costs associated with responding to cat-related complaint calls, impoundment, sheltering, and euthanasia. They underscore the fact that investing in free sterilization is more cost-effective than enforcement-driven approaches [4], which increase the long-term management costs. While financial investment is essential, the success of a Community Cat Program equally depends on community collaboration, trust, stakeholder engagement, and strategic delivery—factors that not only ensure program effectiveness but also generate significant savings. Over time, as the program becomes established and sustainable, the associated costs typically decline. These assistive programs effectively reduce the numbers of unsterilized free-roaming cats and thereby reduce the risk of predation, spread of disease, and nuisance behaviors.

Multiple factors contribute to effective urban cat management. Primarily, the most critical factor is understanding the community’s needs and gaining their trust while ensuring that council employees and AMOs embrace an assistive approach rather than the traditional enforcement-based approach, including engagement with updated training. Secondly, a high-intensity, targeted, and microtargeted cat sterilization program that is free of cost to the community, with strategized accessibility and the removal of known barriers, is necessary. Thirdly, knowledge of the underlying factors contributing to the historical and perpetual failure of traditional cat management is required at all levels of government for effective management programs to be implemented, funded at a sufficient magnitude, and sustained. Lastly, legislative changes at both the state and local government levels are required to make these programs more effective in reducing free-roaming cats and associated issues. Critically important is the classification of semi-owned and unowned cats living in areas where people live and frequent as domestic cats so that they can be managed by non-lethal methods, recognizing the value of cats in people’s lives. This approach is aligned with a One Welfare strategy that benefits human and animal well-being and their physical and social environments.

## Data Availability

Relevant data are reproduced in the text.

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
