# Peer review of "Urban Cat Management in Australia—Evidence-Based Strategies for Success"

_animals, 2025, doi:10.3390/ani15081083_

Round 1

Reviewer 1 Report

Comments and Suggestions for Authors

The paper and its results are outstanding and it adds significantly to the literature.  The only "problems" I have with the paper generally, are related to organization.  I got a bit confused by the various sections and subsections that did not seem to track well.  Here are some specific comments.  Thank you for the oppportunity to review.

(1) Page 1:  Perhaps add an Introduction that lays out the flow of the paper, highlights the key takeaways and explains to the reader what to expect as a guide.

(2) Page 1:  The reference to definition of "feral" cat and distinguishing it from that in the US is a bit confusing.  Be clear that the US does not distinguish the definition of cats based on their proximity or reliance on humans unlike Australia.  Also, there seems to be a concern that the cats involved in these programs might be defined as "feral" by Australizn authorities.  If so, that seems hard to understand as clearly they live among and are reliant on humans.  If this is a theme here make it more explicit and discuss it.  Otherwise perhaps do not mention it at all after defining the terms.

(3) Page 2:  When describing the legal regime note the laws that detract from effective cat management, e.g. cat limits, outlawing TNR/RTF, etc. so the reader is informed of these before discussion of them later.

(4) Section 3 should be better titled as summarizng the three new cat management programs with subsections for each one, e.g. Bayule, NSW and Qld, rather than have a subsection titled "Models of Community Cat Program" 

(5) Section 4 seems designed to summarize the results (previously reported in other papers) of the three programs.  Make this more explicit in the title of Section 4 and the first sentence or two.  Rather than merely state "We have previously reported", be clear to state that the three programs studied have been previously described and reported in the other studies and this section will summarize the results.

(6) Section 4 at page 7 arguably should be subsection 4.7, no? This section lays out the organization of the remainder of the paper, I think.  If I am correct, then make that explicit and tie the description of each of the four key/critical succes factors here directly to the each of the next four major sections (see my comments below).

(7) The first factor here focuses on "people" and thus under subsection 4.7 the first factor should mention "people" when describing community needs, etc. 

(8) Section 5.5, I might reverse the two paragraphs.

(9) Section 6: Perhaps title should state: Second Key Factor . . . just to demonstrate you are working through the four factors descibed above. I note that you sometimes use Key and other times Critical to describe the factors, I am not sure if the distinction is important. If not, perhaps keep the terminology consistent.  If it is important, perhaps explain the difference...?

(10) The numbering of Section 6 should be consistent with other subsections, 6.1, 6.2, etc.  NOT 6.1.1, 6.1.2, 

(11)  The mention of subsidizing veterinary practices for euthanasia is abrupt and perhaps should be introduced, e.g provided context, just briefly.

(12)  Section on Sustainability I think should be a new Section 7 and  tied back to the list of factors mentioned in Section 4.  Also, it would be helpful to explain a bit more why the programs changed so dramatically with the loss of certain personnel in 2021 especially if it was financially so beneficial as mentioned under Section 4.4. 

(13) Section on Need for leg change I think should be new Section 8 and again tied back to the list of factors mentioned in Section 4.  This Section should have subsections devoted to the differing law mentioned, e.g. mandatory ownership, TNR/RTF, Cat Limits, etc.  

(14) Line 881 appears to begin a new section that summarizes the various legal changes recommended.

(15) The subsection on TNR/RTF and its use in the US should include some discussion as to how allowing such programs would affect the three studies discussed.  Arguably there would be greater compliance if folks did not have to own or semi-own the cats....?  but also it might result with more cats without folks to care for them....?  The mention of reuniting cats with their owners is a bit "misleading." Most TNR/RTF in the US is for unowned cats, not friendly cats who get to go outdoors.  Of course if those cats are impounded, that is problematic and their owners will not likely find them, BUT is that scenario the main thrust of this paper?  If so, perhaps make that clearer up front.

(16) The Conclusion is now Section 9 (if you follow my prior comments re: organization), and perhaps emphasize more the financial benefits of these programs as mentiond in Section 4.4.   The reduced cost overall for the pound seems quite compelling and goes directly to the interests/concerns of the government.

Reviewer 2 Report

Comments and Suggestions for Authors

Review comments

General comments

The manuscript titled ‘Urban cat management – critical success factors’ makes an important and valuable contribution to understanding effective and ethical approaches to cat management in Australia, with lessons for international audiences. The authors have provided a very detailed and informative account of the Assistive-Based Cat Management approach which they describe through the experience of three Community Cat Programs in Australia. The work takes a broad view of the role, successes and challenges, of these cat management programs and provides many useful insights about their implementation. This work provides a refreshing and timely contribution to literature on cat management in Australia.

This work will be informative and useful to discussions of cat management and provides applied insights into positive rather than punitive approaches to managing cats.

I particular appreciate the authors:

·       Detailed description of benefits, success factors, and barriers was very useful. I also appreciate that the first benefits described were to cats themselves. Too often the animals we write about become abstracted, it’s nice when they are acknowledged.

·       thorough and clear highlighting of  how AMOs can take on an assistive role in the community and the benefits this has, compared to punitive enforcement approaches. It was nice to see the discussion of how various community support services, e.g., social support services, can collaborate with AMOs and play a positive role in community cat programs.

·       Provide a frank, update about the challenges in the sustainability of Assistive-Based Cat Management programs.

I think this is a valuable and high-quality piece of work and should be published. I provide the following feedback to highlight potential opportunities for clarification or improvement, though respect the authors choice to parse these suggestions based on their own expertise and judgement.

1.       Somewhere early on it would be helpful to include a short paragraph providing context about what the paper is doing and the authors experience in relation to the programs and issue being discussed. Clearly some, if not all, authors have a close relationship with one or more of the community cat programs. Knowing that this piece is a summary of insights gained through over a decade of direct experience with AMOs and Assistive-Based Cat Management would reinforce the authority with which the authors make their claims and present this information.

2.       Throughout the manuscript the authors make claims that enforcement methods are ineffective and indeed harmful to human and animal stakeholders. I would have liked a clearer argument early on that lays out why enforcement approaches are not desirable. This should include reference to evidence that highlights the ineffectiveness of traditional enforcement methods. While I know this is implied in the results presented from Banyule and the other two locations, I think the many might be dismissive because they have a different goal. For many people in Australia the goal is to have no free-roaming cats at all. This is why we see lobbying for 24 hr cat curfews across the country (and the option of impounding and potentially killing those not kept inside by ‘responsible cat owners’). I think it could be useful to address this criticism. It could be as simple as recognising a difference in ‘success criteria’. Where the authors here seem to have a very cat and human centered wellbeing criteria, as well as criteria related to reduction in cat impoundments, complaints, costs, increased community trust, and reduced cat numbers. All of which are clearly legitimate but are often correlated with a certain set of values. Many others in Australia with a different set of values would instead have success criteria related to attempts at total removal of free-roaming cats from our communities. While this is arguably unrealistic it doesn’t stop the goal from being set and in fact many councils around the country are bringing in rules attempting to achieve this. People with such positions might dismiss outright approaches that recognise a place for free-roaming cats in our communities and environments. I’m not sure there is an answer to this issue of opposing values positions but I think it could be good to briefly acknowledge the challenge.

And related, Ln 90-91: “…which has failed over decades to resolve the issue of free-roaming cats in urban areas” – Is there any specific data that could emphasise this point, e.g., the persistent number of complaints and cats being detained/killed.

Line comments

Ln 21 – add space after ‘urination,’

Ln 346 – remove 0 after urbs0 at start of line.

Ln 347-351 – I think this claim is fairly speculative beyond simply saying fewer cats will likely result in wildlife benefits, the direct evidence is unclear for its benefit to urban parkland specifically. For instance, Lilith et al (2010) showed that changes in cat management regimes didn’t impact biodiversity in a local parkland. Fardell et al (2021) show that cats don’t tend to have a preference for vegetated areas over urban areas.

·       Lilith, M., Calver, M., Garkaklis, M., 2010. Do cat restrictions lead to increased species diversity or abundance of small and medium-sized mammals in remnant urban bushland? Pac. Conserv. Biol. 16, 162–172. https://doi.org/10.1071/pc100162

·       Fardell, L.L., Young, L.I., Pavey, C.R., Dickman, C.R., 2021. Habitat use by wandering pet cats (Felis catus) in a patchy urban environment. Journal of Urban Ecology 7, juab019. https://doi.org/10.1093/jue/juab019

Ln 371-373 – I’d like more detail on this in the introduction. I think the evidence within the benefits section of this manuscript are clear, in regard to the wellbeing and financial benefits of Assistive-Based Cat Management, but I don’t think the argument against traditional methods is clear enough. Especially for many who hold different values towards cats and see cat deaths and the consequences of that, e.g. human mental health impacts, as a necessary evil to achieve environments free of cats. I think the MS would benefit greatly from including a stronger and evidence-based case against traditional management approaches being made in the introduction.

Ln 427-430 – This sentence is a little confusing, I think it’s the double up of ‘it is when’ and ‘this is when’.

Ln 538-542 – I think the text connecting these two sentences needs clarifying, it’s unclear how the first sentences enables the conclusion that the training process is applied to nearly every AMO. Instead, do you mean that a standard training should be provided to all AMOs?

Ln 684 – change to ‘4.1 cats/1000 residents’ as is done a few lines later.

Reviewer 3 Report

Comments and Suggestions for Authors

The following review article examines the community cat program in Australia and its impact on shelters from a one-world perspective. It is very well written and thorough but very specific to Australia. There is little comparison to similar programs elsewhere. If the authors wish to keep the focus of the review on Australia, I would suggest changing the title to reflect that. 

Second, the review is very long and redundant in places, so some additional editing would improve the paper. Lastly, there are a few places where the authors report information without reference to data or citations (e.g., describing feral in America Line 84; paragraph lines 256-263). Also, some of the language appears to come from personal experience or interpretation rather than from past studies (e.g., lines 363-377), so I ask the authors to make sure it is evident when the statements come from data/previous research or their interpretation/views.
